# Characterization of Gut Microbiome in the Mud Snail *Cipangopaludina cathayensis* in Response to High-Temperature Stress

**DOI:** 10.3390/ani12182361

**Published:** 2022-09-09

**Authors:** Yang-Yang Wu, Chun-Xing Cheng, Liu Yang, Quan-Qing Ye, Wen-Hong Li, Jiao-Yun Jiang

**Affiliations:** 1Key Laboratory of Ecology of Rare and Endangered Species and Environmental Protection, Guangxi Normal University, Ministry of Education, Guilin 541004, China; 2College of Animal Science and Technology, Guangxi University, 100 Daxue Road, Nanning 530004, China

**Keywords:** thermal stress, *Cipangopaludina Cathayensis*, gut microbiota, high-throughput sequencing

## Abstract

**Simple Summary:**

This study investigated the effects of high-temperature stress on the intestinal microbiome of *Cipangopaludina cathayensis*. High-temperature exposure significantly changed the intestinal microbiota structure of *C. cathayensis*. The relative abundance of putatively beneficial bacteria decreased, whereas the relative abundance of putatively pathogenic bacteria increased after thermal stress. Consistent with the trends of change in the intestinal microbiota, the high-temperature treatment inhibited some carbohydrate metabolism pathways and induced certain disease-related pathways. Thermal stress disrupts the homeostasis of gut microbiota, which may lead to disease outbreak in *C. cathayensis*.

**Abstract:**

The mud snail *Cipangopaludina cathayensis* is a widely distributed species in China. Particularly in Guangxi province, mud snail farming contributes significantly to the economic development. However, global warming in recent decades poses a serious threat to global aquaculture production. The rising water temperature is harmful to aquatic animals. The present study explored the effects of high temperature on the intestinal microbiota of *C. cathayensis*. Snail intestinal samples were collected from the control and high-temperature groups on days 3 and 7 to determine the gut microbiota composition and diversity. Gut bacterial community composition was investigated using high-throughput sequencing of the V3–V4 region of bacterial 16S rRNA genes. Our results suggested that thermal stress altered the gut microbiome structure of *C. cathayensis*. At the phylum level, Proteobacteria, Bacteroidetes, and Firmicutes were dominant in *C. cathayensis* gut microbiota. The T2 treatment (32 ± 1 °C, day 7) significantly decreased the relative abundance of Firmicutes, Actinobacteria, and Deinococcus-Thermus. In T2, the abundance of several genera of putatively beneficial bacteria (*Pseudomonas*, *Aeromonas*, *Rhodobacter*, and *Bacteroides*) decreased, whereas the abundance of *Halomonas*—a pathogenic bacterial genus—increased. The functional prediction results indicated that T2 treatment inhibited some carbohydrate metabolism pathways and induced certain disease-related pathways (e.g., those related to systemic lupus erythematosus, *Vibrio cholerae* infection, hypertrophic cardiomyopathy, and shigellosis). Thus, high temperature profoundly affected the community structure and function of *C. cathayensis* gut microbiota. The results provide insights into the mechanisms associated with response of *C. cathayensis* intestinal microbiota to global warming.

## 1. Introduction

Temperature is an important environmental factor that influences the survival and growth of aquatic organisms [1,2]. In recent decades, human activities and the burning of fossil fuels have increased the level of atmospheric CO_2_, which is estimated to result in an increase of 0.9–5.4 °C in the global mean surface temperatures by the end of the century [3]. Indeed, global warming poses a serious threat to global aquaculture production [4]. Rising water temperature has a series of deleterious effects on the immunity [2,5], metabolism [6,7], reproduction and development [8,9], and behavioral patterns [10,11] of aquatic organisms.

The mud snail *Cipangopaludina cathayensis* (phylum Mollusca, Gastropoda, Prosobranchia, Mesogastropoda, Viviparidae) [12] is a widely distributed species in rivers, lakes, ponds, and other water bodies in China. *C. cathayensis* has high nutritional value, is rich in protein, and contains a balanced proportion of amino acids, particularly umami amino acids [13]. In addition, *C. cathayensis* contains active fibrinolytic proteins [14] and polysaccharides [15] that have biological activity and thus may be used in tumor suppression [14,15]. Indeed, *C. cathayensis* can be used not only as a food source but also for human disease prevention and treatment [14]. *C. cathayensis* is one of the most common cultured aquatic animals in China [13]. In particular, in Guangxi province, mud snail farming plays an important role in the economic development of Guangxi, as mud snails in the family Viviparidae are key components of a popular snack “snail rice noodles” [13]. In recent years, with the rapid increase in the consumption of snail rice noodles, the demand for mud snails has increased [16].

Freshwater ecosystems are particularly fragile under the scenario of global change [17]. Previous studies have found that high-temperature stress has many adverse effects on freshwater mollusks [5,18,19,20,21] and is a key factor that causes mass mortality [21]. To date, few studies have focused on the adverse impacts of thermal stress on *C. cathayensis* [22]. Temperature is a crucial factor affecting the metabolism in *C. cathayensis* [22], and data show that 25 °C is the optimal temperature for *C. cathayensis*. The increase in average water temperatures greatly increases the likelihood of exposure of mud snails to high-temperature stress. As *C. cathayensis* is one of the most important aquaculture species, it is crucial to investigate the effect of thermal stress on *C. cathayensis*.

Gut bacterial communities are important to the host and are considered to constitute an additional organ [23]. The gut microbiome of aquatic animals not only contributes to digestion but also affects nutrition, growth, reproduction, immunity, and vulnerability of the host to diseases [24,25]. Intestinal microbiome homeostasis plays an important role in the health of aquatic animals [26]. Previous studies have shown that thermal stress leads to an imbalance of intestinal microflora [27], provides an ecological niche for invasion by pathogenic flora [28,29,30,31], and increases the risk of pathogen infection [32]. However, the effects of high-temperature stress on the intestinal microbiota of *C. cathayensis* remain unclear. To address this gap, in this study, the effect of high-temperature stress on the gut microbiota composition and diversity in *C. cathayensis* was investigated by exposing *C. cathayensis* over 7 days to two different temperatures (control, 25 ± 1 °C; high temperature, 32 ± 1 °C) [22].

## 2. Materials and Methods

### 2.1. Snail Preparation and Experimental Design

In total, 240 adult snails (*C. cathayensis*) were obtained from Juhe agricultural development cooperatives (25.75909N, 109.386421E), Sanjiang District, Liuzhou City, China. The snails were immediately transported to the laboratory and acclimated in four 50 L tanks (65 × 41 × 20 cm) aerated with an air pump (ACO-318, Hailea, Guangzhou, Guangdong, China) for at least 14 days. The pH value and dissolved oxygen level were 7.5 ± 0.5 and 6.0 ± 0.5 mg/L, respectively [33]. During the acclimation period, specimens were fed with commercial ground fish food (Tongwei, Chengdu, China) once a day at 0.5% of their body weight [34]. The tank water was partially (30%) replaced every day by adding aerated tap water [34].

After acclimation, 180 snails were randomly divided into two groups at 25 ± 1 °C (control group, C) and 32 ± 1 °C (high-temperature group, T). For the treatment group at 32 ± 1 °C, the temperature was gradually increased by 1 °C/day from 25 ± 1 to 32 ± 1 °C to minimize heat shock [29]. Each group had three replicates (30 snails per tank, random allocation). Other conditions in this experiment were consistent with those in acclimation. During the experimental period, one-third of the water in the tank was replaced every day by adding aerated tap water. 

### 2.2. Sample Collection

In total, 36 snails’ intestinal samples were collected from the control and high-temperature groups on day 3 (C1 and T1, nine snails, respectively) and day 7 (C2 and T2, nine snails, respectively) to determine the gut microbiota composition and diversity. The intestines of three snails were pooled into a single sample to ensure sample adequacy, and three biological replicates were established per group. In brief, the shells of sampled snails were wiped with 75% ethanol before the snails were removed from the shell. Then, the snails were dissected and the guts were extracted and rinsed with sterile water three times. The gut samples were flash-frozen in liquid nitrogen and stored at 80 °C for subsequent use.

### 2.3. DNA Extraction, Bacterial 16S rRNA Amplification, and Sequencing

Total genomic DNA (gDNA) of the gut microbiota was extracted using a Fast DNA SPIN Extraction Kit (MP Biomedicals, Santa Ana, CA, USA) following the manufacturer’s protocol. The V3–V4 regions of bacterial 16S rRNA genes were amplified with polymerase chain reaction (PCR) using universal bacterial primers (338F: 5′-ACTCCTACGGGAGGGAGCA-3′, 806R: 5′-GGACTACHVGGGTWTCTAAT-3′). The PCR thermal cycling conditions for each sample were as follows: an initial denaturation at 95 °C for 5 min; 25 cycles of denaturation at 95 °C for 30 s, annealing at 55 °C for 30 s, and extension at 72 °C for 30 s; and then a final extension at 72 °C for 5 min. PCR products were purified and quantified using an AxyPrep DNA Gel Extraction Kit (Axygen, New York, CA, USA) and a Quant-iT PicoGreen dsDNA Assay Kit (Invitrogen, Waltham, MA, USA), respectively. After the individual quantification step, amplicons were pooled in equal amounts, and pair-end 2 × 250 bp sequencing was performed using the Illlumina NovaSeq platform with NovaSeq 6000 SP Reagent Kit (500 cycles) at Shanghai Personal Biotechnology Co., Ltd. (Shanghai, China). The raw reads were deposited in the NCBI Sequence Read Archive database (PRJNA847054).

### 2.4. Sequencing Data Processing

Quantitative Insights into Microbial Ecology (QIIME2, 2019.4) [35] was employed to process the sequencing data. Briefly, raw sequence data were demultiplexed using the demux plugin following by primers cutting with cutadapt plugin [36]. Sequences were then quality filtered, denoised, merged, and chimera removed using the DADA2 plugin [37]. Non-singleton amplicon sequence variants (ASVs) were aligned with mafft [38] and used to construct a phylogeny with fasttree2 [39]. Taxonomy was assigned to ASVs using the classify-sklearn naive Bayes taxonomy classifier in feature-classifier plugin against the Greengenes database (Rlease 13.8, http://greengenes.secondgenome.com/) (accessed on 27 February 2022).

### 2.5. Data Analysis

All steps of sequencing data analysis were performed using QIIME2 and R packages (version 3.2.0). The rarefaction curve was generated by ASVs at a 97% similarity cut-off level. For alpha diversity analyses, Good’s coverage, Shannon diversity, and Simpson indexes were calculated using QIIME2 (for calculation methods, see http://scikit-bio.org/docs/latest/generated/skbio.diversity.alpha.html#module-skbio.diversity.alpha) (accessed on 27 February 2022), and visualized as box plots. Beta diversity was calculated using a weighted Bray–Curtis distance matrix and visualized with principal coordinates analysis (PCoA). Hierarchical clustering using Jaccard distances based on the relative abundances of species was performed to cluster the data set. A Venn diagram was generated to visualize the shared and unique ASVs among samples or groups using R package “VennDiagram”, based on the occurrence of ASVs across samples/groups regardless of their relative abundance. The significance of differentiation of microbiota structure among groups was assessed using PERMANOVA (permutational multivariate analysis of variance) [40], ANOSIM (analysis of similarities) [41], Permdisp [42] using QIIME2. PICRUSt2 (Phylogenetic Investigation of Communities by Reconstruction of Unobserved States; https://github.com/picrust/picrust2) (accessed on 2 March 2022) was used for functional profiling of the intestinal microflora. In brief, Gappa was applied for phylogenetic placement of reads, Castor for hidden state prediction, and MinPath for pathway inference. The Kyoto Encyclopedia of Genes and Genomes (KEGG) pathway database (http://www.genome.jp/kegg/pathway.html) (accessed on 2 March 2022) was used to analyze the differential expression of gut microbial functional pathways under thermal stress. For this, the PICRUSt2 output file pred_metagenome_unstrat.tsv was unzipped and then combined with the appropriate KEGG file using the function inner_join (from dplyr). Differences between populations were analyzed using a *t*-test (SPSS 23.0). Statistical significance was accepted at *p* < 0.05. Data were expressed as mean ± standard deviation (SD).

## 3. Results

### 3.1. High-Throughput Sequencing Data

The microbial communities of *C. cathayensis* were determined using 16S rRNA (V3–V4 region) sequencing analysis. In total, 1,392,723 raw sequences were obtained from the 12 samples, and 1,041,438 effective sequences were obtained after quality processing (Appendix A). In total, 19,225 ASVs were derived from all samples; control treatment groups C1 (day 3) and C2 (day 7) and high-temperature treatment groups T1 (day 3) and T2 (day 7) had 4664, 3260, 6589, and 4712 ASVs per sample, respectively. In addition, 297 ASVs were common among the four treatment groups, whereas 2164, 1322, 3204, and 2004 ASVs were unique to the C1, C2, T1, and T2 treatments, respectively (Figure 1). Rarefaction curves showed that the curves reached a plateau as the number of identified sequences increased, suggesting that the identified sequences could sufficiently cover the bacteria in samples and could be used for further analysis (Appendix A).

### 3.2. Diversity of Intestinal Microflora

Alpha and beta diversities were estimated across different groups to evaluate the effects of high-temperature stress on the structure of *C. cathayensis* intestinal microflora. There was no significant difference in the Shannon and Simpson indexes between the C1 and T1 groups (Figure 2A,B). However, with the extension of the treatment time to 7 days, the Shannon and Simpson indexes decreased significantly in the T2 group compared with the C2 group (*p* < 0.05). These two alpha-diversity indices were also lower (although not significantly lower) in the T2 group than in the T1 group (Figure 2A,B; Appendix A). Similarly, the beta diversity analysis (PCoA based on the Bray–Curtis distance matrix) showed a clear separation in bacterial community composition between high-temperature stress and control treatments (excluding one control sample), with the following main principal component (PC) scores: PC1 = 52.5%, PC2 = 21.3% (Figure 2C). Apparently, T1 and C1 could be separated by PC1, and T2 and C2 could be separated by PC2. Moreover, the results of the PERMANOVA analysis of microbial community structure showed a significant different among all groups (*p* < 0.05, Appendix A). In addition, a hierarchical clustering tree (Appendix A) revealed that microbial communities among the four treatment groups were clustered into two groups, ASVs from the C1 and C2 groups were clustered in one group based on similarity (excluding one control sample), while the T1 and T2 groups clustered into one independent group. Overall, these results indicate that thermal stress significantly affected the diversity of the gut microbiome of *C. cathayensis*.

### 3.3. Effect of High Temperature on the Overall Community Structure of Intestinal Microbiota

In total, 28 phyla, 60 classes, 133 orders, 217 families, 396 genera, and 115 species were identified. At the phylum level, Proteobacteria was the predominant phylum in all intestinal samples (65.04% in C1, 63.62% in C2, 46.26% in T1, and 43.59% in T2). Proteobacteria, Bacteroidetes, Firmicutes, Actinobacteria, and Deinococcus-Thermus together with Chlamydiae accounted for 71.1% of the total abundance of intestinal microbiota (Figure 3A; Appendix A). The relative abundance of Proteobacteria decreased in T1 and T2 groups, however, the difference was not statistically significant (Figure 3B). The relative abundance of Firmicutes was significantly higher in T1 than in C1 (*p* = 0.021; Figure 3C). In contrast, the relative abundance of Firmicutes (*p* = 0.001), Actinobacteria (*p* = 0.001), and Deinococcus-Thermus (*p* = 0.025) was significantly lower in T2 than in C2 (Figure 3C–E). Notably, the Firmicutes/Bacteroidetes ratio in the T2 group was lower than that in the C2 group, but the difference was not statistically significant (Figure 3F).

At the genus level, the dominant genera were *Pseudomonas*, *Acinetobacter*, *Pelomonas*, *Bacteroides*, *Aeromonas*, *Aquabacterium*, *Rhodobacter*, *Halomonas*, *Muribaculaceae*, and *Brevundimonas* and accounted for 41.67% of the total abundance of intestinal microbiota (Figure 4A; Appendix A). With the increase in treatment time, the relative abundance of *Bacteroides* (*p* = 0.027; Figure 4B), *Pseudomonas* (*p* = 0.008; Figure 4C), *Aeromonas* (*p* = 0.001; Figure 4D), and *Rhodobacter* (*p* = 0.008; Figure 4E) was lower in the T2 treatment than in the C2 treatment. Remarkably, the relative abundance of the genus *Bacteroides* significantly increased in the T1 group (*p* = 0.05) and then decreased in the T2 group (*p* = 0.027). In contrast, the relative abundance of *Halomonas* increased in the T2 group (*p*
*=* 0.015; Figure 4F, Appendix A). 

### 3.4. Functional Prediction of the Intestinal Microflora

To further evaluate the function of gut microbiota in response to high-temperature stress in *C. cathayensis*, we predicted the gene function of the gut microbial community. The PICRUSt functional predictions and the KEGG pathway enrichment results showed that six functional groups (pathway level 1) were identified, including metabolism (80.53%), genetic information processing (10.55%), cellular processes (5.63%), environmental information processing (2.80%), human diseases (0.42%), and organismal systems (0.35%; Figure 5). We found that some carbohydrate metabolism pathways were enriched in the T1 treatment compared to the control group (Figure 6A); moreover, one potentially disease-related pathway (bacterial invasion of epithelial cells) was also elevated in the T1 treatment (Figure 6A). Notably, carbohydrate metabolism pathways were found to be inhibited in the T2 group relative to the T1 group (Figure 6C). Furthermore, other disease-related pathways (e.g., those related to systemic lupus erythematosus, *Vibrio cholerae* infection, hypertrophic cardiomyopathy, and shigellosis) were increased in the T2 treatment (Figure 6B,C). Taken together, these findings suggest that thermal exposure affects the metabolism of the gut microbiota and in turn the health of *C. cathayensis*. 

## 4. Discussion

The gut microbiome often serves as an additional organ [23] that plays an important role in nutrient digestion and absorption, immunity, and development of the host [24,25]. Environmental shift is an important abiotic factor affecting the intestinal microflora structure in aquatic animals, including mollusks [28,29,30,32,43]. Indeed, temperature affects the composition and function of mollusk intestinal microbiota [28,29,30,32,43]. Changes in the gut microbiome were found to be an important process through which animal hosts acclimate to temperature shifts [44]. However, the effects of thermal stress on the intestinal microbiota of *C. cathayensis* remained unclear. In the present study, *C. cathayensis* were exposed over 7 days to two different temperatures (control, 25 ± 1 °C; high temperature, 32 ± 1 °C). 16S rRNA gene sequencing technology was used to analyze the effects of high-temperature stress on the intestinal microflora structure and function in *C. cathayensis*. 

In the present study, although there was no significant difference in alpha diversity between C1 and T1, a reduction was observed in the T2 group with increased exposure time. Similarly, previous studies reported decreased alpha diversity in some mollusks [28,30,45] exposed to high-temperature stress. In contrast, some studies detected an increase in the alpha diversity metrics in mollusks [29,43] and crustaceans [46]. Apparently, the response strategies of the intestinal microbiome under thermal stress are related to host taxonomy. Indeed, different microbiome responses to temperature have been observed in closely related species [28,29,30]. In addition, PCoA and hierarchical clustering results showed that the gut microbial community structure in the control groups, excluding one control sample, was distinct from that in the T1 and T2 groups; the latter two treatments also showed significantly different gut microbial community structures from each other, implying that high-temperature exposure significantly changed the intestinal microbiota structure in *C. cathayensis*. Gut microbiota are beneficial for their hosts [24,25]; thus, the decrease in alpha diversity and changes in the gut microbial community structure following high-temperature exposure observed in the present study may have adverse effects on *C. cathayensis* [29,30].

In this study, Proteobacteria, Bacteroidetes, and Firmicutes were the three dominant phyla across all treatment groups, which is consistent with previous findings in mollusks [47,48]. However, Bacteroidetes and Firmicutes were not the dominant phyla in a closely related species, *Cipangopaludina chinensis* [48,49]. This inconsistency implies that although the two species are closely related, they may have different strategies of responding to high-temperature stress [28,29,30]. Proteobacteria, widely used as probiotics in aquaculture [50], provide many benefits to aquaculture animals [50,51,52]. In the present study, the relative abundance of Proteobacteria slightly decreased in the T1 and T2 treatment groups, suggesting that high-temperature exposure has adverse effects on *C. cathayensis* health [43,46]. 

Studies have shown that Firmicutes plays an important role in maintaining the gut homeostasis [53] and actively responds to temperature changes [54]. Notably, in the present study, the relative abundance of Firmicutes increased significantly in T1 and the decreased in T2, suggesting that thermal stress affects the gut homeostasis of *C. cathayensis* [54]. In addition, Firmicutes is known to be positively correlated with intestinal nutrient metabolism [32,46], and a reduction in Firmicutes abundance may hinder intestinal nutrient absorption [32]. It is also worth noting that the Firmicutes/Bacteroidetes ratio slightly decreased in T1 and T2. The Firmicutes/Bacteroidetes ratio is an important biomarker of gut dysbiosis [55]; a decreased Firmicutes/Bacteroidetes ratio affects nutrient metabolism and restricts the growth of aquaculture animals [32,56]. Thus, a decrease in the Firmicutes/Bacteroidetes ratio may have adverse effects on *C. cathayensis* growth [56].

Similarly, Actinobacteria and Deinococcus-Thermus were both found to be decreased in our study. Actinobacteria, widely present in soil, freshwater, and marine ecosystems and found in plants and animals [57,58,59], are endowed with the ability to biosynthesize and produce secondary metabolites that play a vital role in nutrient recycling and regulation of a wide variety of diseases [60]. However, the abundance of Actinobacteria decreased significantly in T2, implying that the synthesis of bioactive compounds is disrupted under high-temperature stress, potentially harming the host. The phylum Deinococcus-Thermus comprises several thermophilic species, which are highly tolerant to high temperatures [61], and their relative abundance in mussel *Mytilus coruscus* showed significant differences in response to temperature [28]. In addition, this phylum was also found to include extremophiles, which are associated with the production of carotenoids [62] and exhibit resistance to extreme environmental stressors such as radiation, oxidation, and heavy metals [62,63]. A decreased relative abundance of Deinococcus-Thermus in the present study suggests a decreased adaptation potential to high temperature.

At the genus level, the abundance of *Pseudomonas*, *Aeromonas*, *Rhodobacter*, *Bacteroides*, and *Halomonas* was found to be significantly different among different treatments. *Pseudomonas* is widely distributed in the gut of aquatic animals [25,64] and has been identified as pathogenic in teleosts [25,65]. However, *Pseudomonas* is also used as a probiotic in aquaculture [66]. In the present study, the relative abundance of *Pseudomonas* significantly decreased in the T2 treatment, implying that thermal stress exerts adverse effects on this genus. Further research is required to verify whether *Pseudomonas* acts as a probiotic in *C. cathayensis*. Similarly, the relative abundance of *Aeromonas*—which is potentially beneficial to aquatic animals in many aspects, including growth, immunity, and disease resistance—was decreased in T2 [52,67], suggesting that high temperatures disrupt the homeostatic balance of probiotics and adversely affect the health of the host. Moreover, the abundance of *Rhodobacter* also decreased significantly in T2. *Rhodobacter* is a candidate probiotic for fish [68], which can improve growth, reduce oxidative damage, and stimulate the immune system [69]. Decreased *Rhodobacter* abundance has been reported to reduce growth [70] and have adverse effects on fish innate immunity [71], resulting in increased vulnerability to disease [72]. *Bacteroides* species are highly adapted to the gut environment, and they live and grow exclusively in the gastrointestinal tracts of mammals [73] and aquatic animals [74]. Although members of this group are known to be opportunistic pathogens [28], most of them actively enhance the gut environment, establish stable and long-term associations with their hosts, and confer numerous health benefits [73]. Notably, *Bacteroides* release commensal molecules (e.g., polysaccharide A and sphingolipids) that play an important role in maintaining intestinal homeostasis and modulating the host immune system in health and disease [75,76]. A reduction in the abundance of *Bacteroides* has been associated with inflammatory bowel disease [77]. In our study, the relative abundance of *Bacteroides* significantly increased in T1 and decreased in T2, implying that *Bacteroides* respond positively to high temperatures; however, extended thermal stress disrupts the homeostasis of *Bacteroides*, which may lead to disease outbreak in *C. cathayensis* [77]. 

In contrast, under high-temperature stress, the abundance of *Halomonas* was elevated in T2. *Halomonas* species, commonly found in saline environments, are halotolerant or halophilic [78] and cause infections in humans [79]. *Halomonas* strains are also considered to be pathogenic in fish [80,81]. Indeed, *Halomonas* are found to be elevated in the intestine of zebrafish (*Danio rerio*) after pathogen invasion [80] or exposure to environmental stress (e.g., *Microcystis aeruginosa* and phthalate esters), inducing intestinal innate immune responses [80], which results in intestinal injury [82]. These findings illustrate that the health of *C. cathayensis* exposed to thermal stress may be impaired.

To further assess the effect of high temperature on *C. cathayensis*, we examined the functional pathways encoded by *C. cathayensis* gut microbiota. The functional prediction analysis showed that most genes of the *C. cathayensis* gut microbiota were related to metabolism. Gut microbiota contributes to host nutrition, absorption, and metabolism [54]; thus, thermal stress may affect *C. cathayensis* metabolism [54]. Indeed, in the present study, some carbohydrate metabolism pathways were found to be enriched in the T1 group but decreased in the T2 group compared to the T1 group. These results are consistent with the change trend of Firmicutes abundance observed in different treatment groups, which is known to be positively correlated with intestinal nutrient metabolism and nutrient absorption [32]. A reduction in the abundance of Firmicutes was reported to hamper growth in aquaculture animals [32,56]. Therefore, we speculate that increasing temperatures disrupt Firmicutes homeostasis and inhibit carbohydrate metabolism pathways, thus, inhibiting the growth of *C. cathayensis*. Gut microbiota homeostasis also plays an important role in host health [80,82]. Under thermal stress, one disease-related pathway was found to be enriched in T1, whereas more disease-related pathways were detected in T2. This finding is consistent with the results that thermal stress significantly decreased the abundance of some functionally important taxa including the phyla Firmicutes, Actinobacteria, Deinococcus-Thermus and the genera *Pseudomonas*, *Aeromonas*, *Rhodobacter*, and *Bacteroides* and increased the abundance of *Halomonas*, a potentially pathogenic genus, in T2. The functional changes in the intestinal microflora are associated with microflora composition in mud snails under high-temperature stress and the deficiency of functionally important gut microbial taxa may increase their susceptibility to disease [32].

## 5. Conclusions

Our results revealed that the structure and function of intestinal microbial communities in *C. cathayensis* differed between the control treatment and high-temperature treatment. High-temperature stress disrupted the homeostasis of the intestinal microbiota of *C. cathayensis*, as it decreased the abundance of putatively beneficial bacteria and increased the abundance of putatively pathogenic bacteria. Moreover, thermal stress profoundly influenced the metabolic functions of gut microbiota, which may increase the disease susceptibility of *C. cathayensis*. To the best of our knowledge, this is the first study to explore the effects of high-temperature stress on the intestinal microbiota of *C. cathayensis*. The results obtained in this study provide insights into the mechanisms associated with the response of the intestinal microbiota of *C. cathayensis* to global warming. However, due to the small sample size and obtaining the 16S rRNA gene sequences through the Illumina HiSeq platform our study may have limitations. In the present study, we did not isolate and identify the putatively pathogenic and putatively beneficial bacteria, which warrants further research.

## Figures and Tables

**Figure 1 animals-12-02361-f001:**
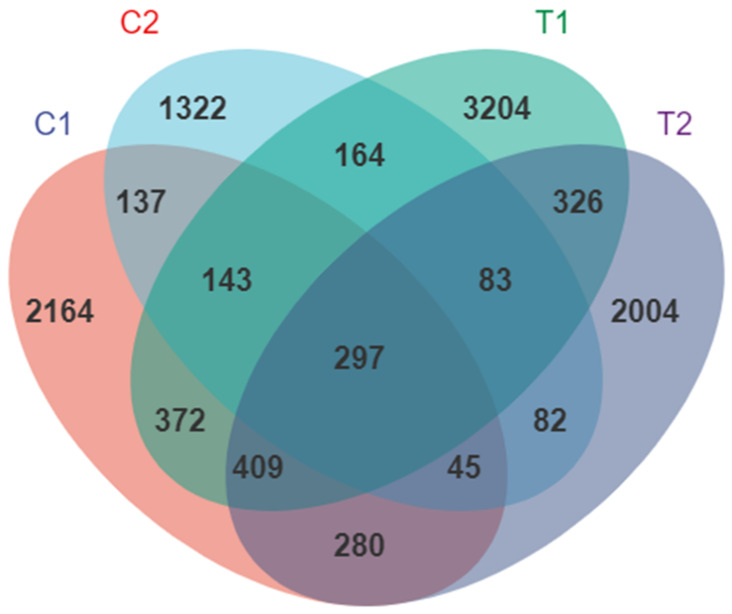
Venn diagram depicting the numbers of shared and unique amplicon sequence variants (ASVs) among *Cipangopaludina cathayensis* treatment groups C1 (control treatment; 25 ± 1 °C, day 3), C2 (control treatment; 25 ± 1 °C, day 7), T1 (high-temperature treatment; 32 ± 1 °C, day 3), and T2 (high-temperature treatment; 32 ± 1 °C, day 7).

**Figure 2 animals-12-02361-f002:**
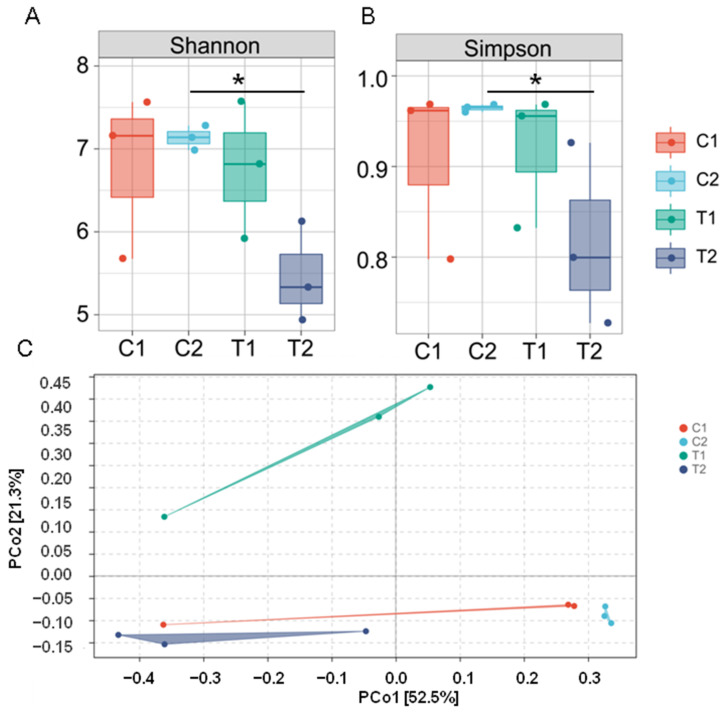
Intestinal microbiome diversity among *Cipangopaludina cathayensis* treatment groups C1 (control treatment; 25 ± 1 °C, day 3), C2 (control treatment; 25 ± 1 °C, day 7), T1 (high-temperature treatment; 32 ± 1 °C, day 3), and T2 (high-temperature treatment; 32 ± 1 °C, day 7). (**A**) Shannon diversity index; (**B**) Simpson diversity index; (**C**) Bray–Curtis distances calculated and visualized through principal coordinates analysis (PCoA; ellipses were drawn with 95% confidence intervals). Asterisks indicate *p*-values < 0.05.

**Figure 3 animals-12-02361-f003:**
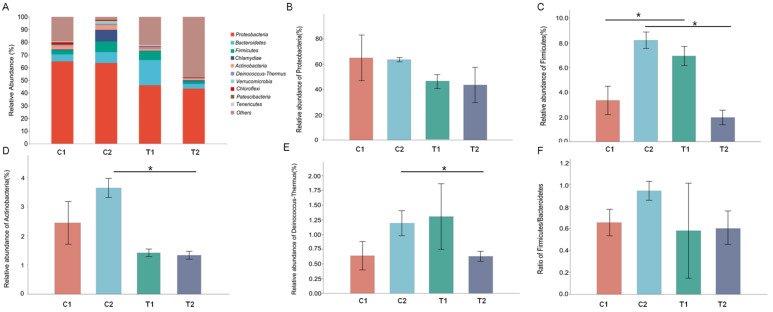
Composition of *Cipangopaludina cathayensis* intestinal microflora at the phylum level. (**A**) Composition of the intestinal microflora among treatment groups C1 (control treatment; 25 ± 1 °C, day 3), C2 (control treatment; 25 ± 1 °C, day 7), T1 (high-temperature treatment; 32 ± 1 °C, day 3), and T2 (high-temperature treatment; 32 ± 1 °C, day 7); (**B**) relative abundance of phylum Proteobacteria; (**C**) relative abundance of phylum Firmicutes; (**D**) relative abundance of phylum Actinobacteria; (**E**) relative abundance of phylum Deinococcus-Thermus; (**F**) Firmicutes/Bacteroidetes ratio. Asterisks indicate *p*-values < 0.05.

**Figure 4 animals-12-02361-f004:**
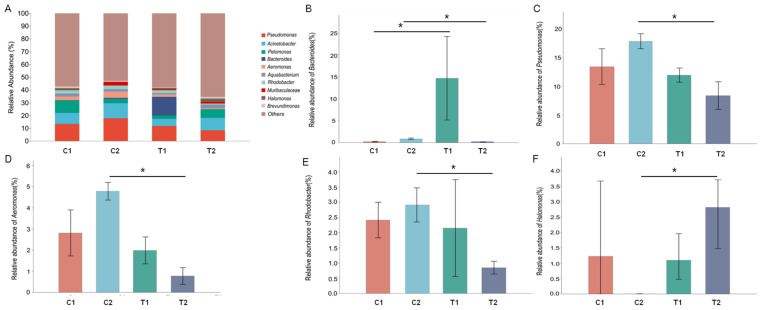
Composition of *Cipangopaludina cathayensis* intestinal microflora at the genus level. (**A**) Composition of the intestinal microflora among treatment groups C1 (control treatment; 25 ± 1 °C, day 3), C2 (control treatment; 25 ± 1 °C, day 7), T1 (high-temperature treatment; 32 ± 1 °C, day 3), and T2 (high-temperature treatment; 32 ± 1 °C, day 7); (**B**) relative abundance of the genus *Bacteroides*; (**C**) relative abundance of the genus *Pseudomonas*; (**D**) relative abundance of the genus *Aeromonas*; (**E**) relative abundance of the genus *Rhodobacter*; (**F**) relative abundance of the genus *Halomonas*. Asterisks indicate *p*-values < 0.05.

**Figure 5 animals-12-02361-f005:**
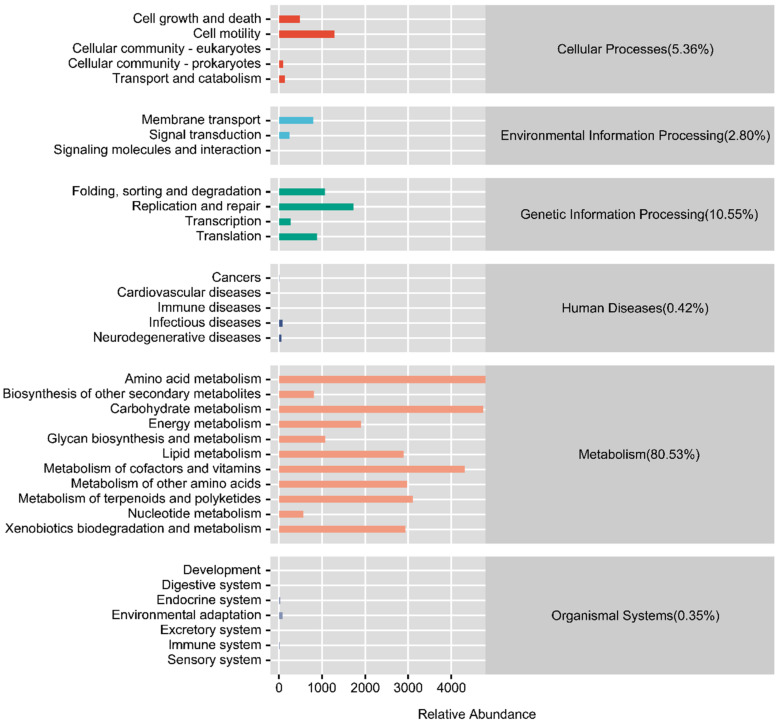
Functional annotation and relative abundance of *Cipangopaludina cathayensis* intestinal microbiota using Kyoto Encyclopedia of Gene and Genomes (KEGG) analysis at level 1 and level 2. The relative abundance of genes associated with a function is shown using colored bars (level 1). The detailed pathways are shown on the left (level 2).

**Figure 6 animals-12-02361-f006:**
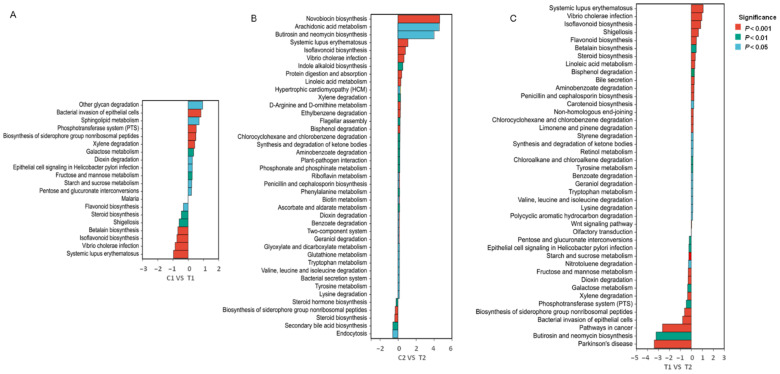
Metabolic functions of *Cipangopaludina cathayensis* intestinal microbiota in all samples predicted on the basis of the Kyoto Encyclopedia of Genes and Genomes (KEGG) database. (**A**) Significantly different pathways between C1 (control treatment; 25 ± 1 °C, day 3) and T1 (high-temperature treatment; 32 ± 1 °C, day 3); (**B**) significantly different pathways between C2 (control treatment; 25 ± 1 °C, day 7) and T2 (high-temperature treatment; 32 ± 1 °C, day 7); (**C**) significantly different pathways between T1 (high-temperature treatment; 32 ± 1 °C, day 3) and T2 (high-temperature treatment; 32 ± 1 °C, day 7).

## Data Availability

The raw sequencing reads that support the findings of the present study are publicly available in the NCBI Sequence Read Archive (SRA) database at (https://www.ncbi.nlm.nih.gov, accessed on 8 June 2022) under accession number PRJNA847054.

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
