# Peer review of "Characterization of Gut Microbiome in the Mud Snail *Cipangopaludina cathayensis* in Response to High-Temperature Stress"

_animals, 2022, doi:10.3390/ani12182361_

Round 1
Reviewer 1 Report
Dear Authors,
Thank you for submitting this interesting study that investigates the gut microbiome of mud snails following differing periods of heat stress. I found your study to be engaging and well considered.
There are some revisions required to ensure the study is scientifically robust, and I have provided specific feedback on a PDF version of the paper. Please pay specific attention to the methods, especially regarding snail management and sampling methods, as there are some gaps in this section that limit repeatability.

Reviewer 2 Report
The authors investigated the effect of temperature stress on the characterization of the gut microbiome in the Cipangopaludina cathayensis. They sampled on two days (day 3 and 7) in two treatments to test their hypothesis. This manuscript (MS) was clearly written and easy to understand. Although they did a great job, a fundamental problem in the experimental design stopped me from reviewing this MS. They analysed only one animal each time which is not never enough and the data from this MS is not defendable and reliable. Therefore, the authors cannot make any conclusions from this data.
Kind regards
Reviewer 3 Report
Throughout the text: please use italics for the species names
Line 53: what do you mean by “a balanced proportion of amino acids,”?
Line 56: please correct the citation (14,15), use brackets []
Line 62: any citation to support the line “the demand for mud snails has increased.”?
Line 88: “The snails were immediately transported to the laboratory and acclimated in 50 L tanks” do you mean that all 240 snails were moved to one tank or were divided to several tanks? If the latter is the case, please provide more details about the numbers. Also, how do you ensure that there would be no overpopulation in the tanks (whatever the number of snails are) that would affect the stress levels of the animals?
Line 90-93: “dissolved oxygen level were 7.5 ± 0.5 and 6.0 ± 0.5 mg/L” please provide citation suggesting that these are the optimum conditions for culture. Same comment for the quantity of food and water replacement rate.
Line 94: “180 snails” what about the other 60 snails? Is there any mortality during the acclimation?
Line 95: why did you choose 32 C? is there any argument that this would be the water temperature in X years from now? Is it the current water temperature somewhere within the species range now? What is the mortality temperature for this species? My concern is how you ensured that this temperature is not too high or unrealistic within a global warming scenario concerning the near future. Also, justify that the increase of 1C/day is not too abrupt or that one day period is not too short as a period. The warming of the waters due to the global warming would be gradual so individuals would have the chance to acclimate to the new conditions, at least up to one point, so the acclimation part is important e.g., the temperature change every one day is it enough time for the species? what if you increase the temperature by e.g., 0.1 C every X days and at some point reach the 32 C? would this have the same effect? In conclusion, I think that it is important to justify the choice of 32 C and the pace of temperature increase you chose.
Line 102: so practically you analysed 3 samples per group? (3 snails pooled into one sample and finally 3 samples per group, right?)
Lines 118-143: please provide more information about the whole process after the PCR. It is not clear for example how you separate the different 16S products after the PCR. You may use (just an idea, not a request) photos or a graph of the process in order to be more descriptive maybe in supplementary material.
Also, more details on the sequencing part are needed. I think that this part in general needs a more descriptive phrasing.
Also, the data processing part needs more descriptive phrasing. For example: “Taxonomy was assigned using the DADA2 pipeline which implements the naive Bayes classifier using the DADA2 default parameters using the Greengenes database” I would recommend you to add 2-3 lines explaining how this method works.
More important, the PICRUSt2/KEGG analysis needs a lot more explanation on how it works and how you implemented it on your data. I would strongly recommend to add 5-6 lines to provide this information.
Also concerning the analysis part, you practically have 3 samples (since pooling was done) per group (control and treated) per sampling (day3 and day7). This is a small sample size. Is this sample size adequate for your analyses? Did you perform a correction for the small sample size? For example, definitely 3 samples are not adequate for performing an ANOVA.
Line 160: “sufficient sequencing depth”, how much? Please add this information
Line 177: “(excluding one control sample),” why this exclusion was necessary? You already had only 3 control samples!
Line 176: “clear separation”. You mean T1-C1 based on PC2 and T2-C2 based on PC1. Please rephrase to be clearer.
Line 268: “different microbiome responses to temperature have been observed in closely related species” please explain
General comment on the conclusion and abstract: please rephrase the parts were mention effect of the thermal stress, actually the prolong thermal stress seem to have an effect on the microbiome since for the 3 days stress most of the analysis did not support that.
My main concern on this study is the small sample size. I am not sure how strong the conclusions could be based on only 3 samples (and at some point you also excluded one sample from the control group).
Round 2
Reviewer 2 Report
The authors improved the quality of the MS. My concern was the sample size of this study which it seems you used three snails. However, it still is not big enough and should be mentioned clearly in the conclusion section as a limitation of your study.
Author Response
Dear reviewer,
We have pointed out the small sample size of this study in our conclusion.
Thank you very much for your valuable advice!
Sincerely
Jiaoyun Jiang